

# Impervious surface and local abiotic conditions influence arthropod communities within urban greenspaces

Garrett M. Maher, Graham A. Johnson and Justin D. Burdine

Division of Science and Kinesiology, Cornerstone University, Grand Rapids, MI, United States of America

## ABSTRACT

The abundance of arthropods is declining globally, and human-modification of natural habitat is a primary driver of these declines. Arthropod declines are concerning because arthropods mediate critical ecosystem functions, and sustained declines may lead to cascading trophic effects. There is growing evidence that properly managed urban environments can provide refugium to arthropods, but few cities have examined arthropods within urban greenspaces to evaluate their management efforts. In this study, we surveyed arthropod communities within a medium-sized, growing city. We investigated arthropod communities (abundance, richness, diversity, community composition) within 16 urban greenspaces across metropolitan Grand Rapids, Michigan (USA). We focused our efforts on urban gardens and pocket prairies, and measured environmental variables at each site. We collected 5,468 individual arthropods that spanned 14 taxonomic orders and 66 morphospecies. The results showed that community composition was influenced by impervious surface, white flower abundance, and humidity. Total arthropod abundance and diversity were positively associated with humidity. For specific orders, Hymenoptera (bees, ants, wasps) abundance was negatively associated with temperature, and positively associated with site perimeter-area ratio. Hemiptera (true bugs) were negatively associated with impervious surface and positively associated with humidity. These findings show that impervious surfaces impact arthropod communities, but many of the observed changes were driven by local abiotic conditions like temperature and humidity. This suggests that management decisions within urban greenspaces are important in determining the structure of arthropod communities. Future studies on arthropods in cities should determine whether manipulating the abiotic conditions of urban greenspaces influences the composition of arthropod communities. These results should inform city planners and homeowners of the need to properly manage urban greenspaces in cities to maintain diverse arthropod assemblages.

Corresponding author
Justin D. Burdine,
justin.burdine@cornerstone.edu

## INTRODUCTION

Urban environments are understood to be regions that have been altered to accommodate a growing human population, and are characterized by high densities of people and impervious surfaces such as roads, buildings, sidewalks, and parking lots. Over half the

global human population resides in urban environments, and 82% of the United States' population is classified as urban (*United Nations, 2018*). The global human population continues to rise, and the United Nations estimates that an additional 2.5 billion people will be urban dwellers by 2050 (2018). This places pressure on urban regions to maintain quality habitat for human well-being, and to sustain the delivery of critical ecosystem services for urban residents. Continued urban expansion encroaches into the peri-urban landscape (*Smidt et al., 2018*), and leads to fragmentation and natural habitat loss. Impervious surfaces replace greenspaces as urban centers expand outward into agricultural and natural lands (*Smidt et al., 2018*), and can alter the abiotic conditions of cities, increasing land surface temperature (*Gaffin et al., 2008*) and reducing water availability (*Jiang, Fu & Weng, 2015*). Therefore, it is important to understand how continued urbanization impacts the biotic communities that perform the ecosystem functions that are essential for human well-being in cities.

Many studies have documented the biotic homogenization or convergence of cities (*McKinney, 2006*; *Wittig & Becker, 2010*; *Groffman et al., 2014*; *Lemoine-Rodríguez, Inostroza & Zepp, 2020*), and impervious surfaces play a significant role in establishing similar patterns of fragmentation and ecosystem structure that underlie this homogenization (*Groffman et al., 2017*). An additional component to this homogenization are the plant communities that become established within urban greenspaces. Non-native plants are often drivers of homogenization (*McKinney, 2004*; *Cubino et al., 2019*) because they establish similar habitats that attract the same types of arthropods in cities (*Knop, 2016*). Although biotic homogenization may increase local species richness, diversity at the regional or global scale tends to decline (*Dar & Reshi, 2014*). This leads to functional homogenization as native, specialist species are replaced by non-native generalists that are unable to perform the same ecosystem functions (*Merckx & Dyck, 2019*). However, there is growing evidence that cities can harbor diverse plant communities with arthropods that counter this homogenization effect (*Pardee & Philpott, 2014*; *Hall et al., 2017*; *Joimel et al., 2019*).

Arthropods are the most diverse phylum of animals on Earth, and they maintain ecosystem functions as pollinators, decomposers, herbivores, and predators, amongst others. Within cities, arthropod communities often experience dissimilar responses. The overall abundance and diversity of arthropods tends to decline as impervious surfaces increase (*Lagucki, Burdine & McCluney, 2017*; *Miles et al., 2019*; *Fenoglio, Rossetti & Videla, 2020*). However, arthropod responses to urbanization are also driven by the abiotic and biotic conditions within urban greenspaces. For instance, urbanization can influence temperature and water availability due to urban heat island effects (*Gaffin et al., 2008*), and altered water drainage and infiltration patterns (*Shuster et al., 2005*). Research within cities has shown that temperature is often a better predictor than impervious surface at explaining shifts in the diversity of specific arthropod groups like true flies (*McGlynn et al., 2019*). And studies have found that increased irrigation in urban regions may have positive impacts on specific arthropods like aphids (*Andrade, Bateman & Kang, 2017*). In terms of the biotic conditions, increasing the availability of vegetation (*Turrini & Knop, 2015*), and principally flowers (*Mody et al., 2020*), can lead to increased arthropod abundance. The

types of flowers present also help explain their positive impacts on bees (*Pardee & Philpott, 2014*; *Burdine & McCluney, 2019b*) and natural enemies (*Dale et al., 2020*).

This study focused on arthropod communities in Grand Rapids, Michigan, USA. Urban land in Grand Rapids increased by 204% between 1992 to 2011, and urban land is projected to increase an additional 35% by 2050 (*Smidt et al., 2018*). The city contains many urban gardens and pocket prairies that harbor native plant species, and may be important habitats for arthropods as studies in nearby cities have shown (*Pardee & Philpott, 2014*). Furthermore, these habitats provide valuable ecosystem services, such as food production and recreation, and have been identified as important public greenspaces for urban development (*Turo & Gardiner, 2019*). Grand Rapids is one of the fastest growing cities in the Midwest (*Sharf, 2018*), and is in a unique position because the urban center is surrounded by natural land cover (*Smidt et al., 2018*). Thus, understanding the structure of arthropod communities can provide context for similar studies in the future as urban expansion continues into these natural habitats.

The primary objective of this study was to survey arthropod communities (abundance, richness, diversity, composition) along an urbanization gradient (impervious surface) within a network of urban gardens and pocket prairies. We measured the environmental conditions (percent impervious surface, site perimeter-area ratio, temperature, humidity, total flowers, white flowers, purple flowers) within each site to better understand drivers of change within arthropod communities, and to identify interventions that land managers can implement to support diverse arthropod assemblages. We expected impervious surface at the landscape scale to be the dominant driver in community structure, and that overall arthropod abundance would decline near the urban center (high impervious surface). For biotic conditions, we expected floral resource availability to be the dominant factor in explaining changes in arthropod richness and diversity.

## MATERIAL AND METHODS

### Site selection

Arthropods were sampled from eight urban gardens and eight pocket prairies across metropolitan Grand Rapids, Michigan, USA (see Fig. 1). Urban gardens were predominately compromised of edible plants, but often contained non-edible wildflowers. The urban garden sites were located on non-residential properties, and functioned as community garden spaces for groups of individuals to manage. Pocket prairies were defined as habitats that have undergone a restoration process through the intentional planting of native prairie plant species. Sites were compiled by first establishing a list of known pocket prairies and urban gardens within a 25 km radius of Grand Rapids City Hall as a proxy of the city center. Then, permissions were requested for each site to visit and collect specimens during the summer of 2020. We secured permissions from 16 collection sites that were spatially separated by a minimum distance of 750 m between each site to maintain the spatial independence of each research site and reduce pseudoreplication. Sites were managed by the City of Grand Rapids, City of Hudsonville, Township of Ada, Township of Gaines, Creston Neighborhood Association, Blandford Nature Center, New City Neighbors, Urban

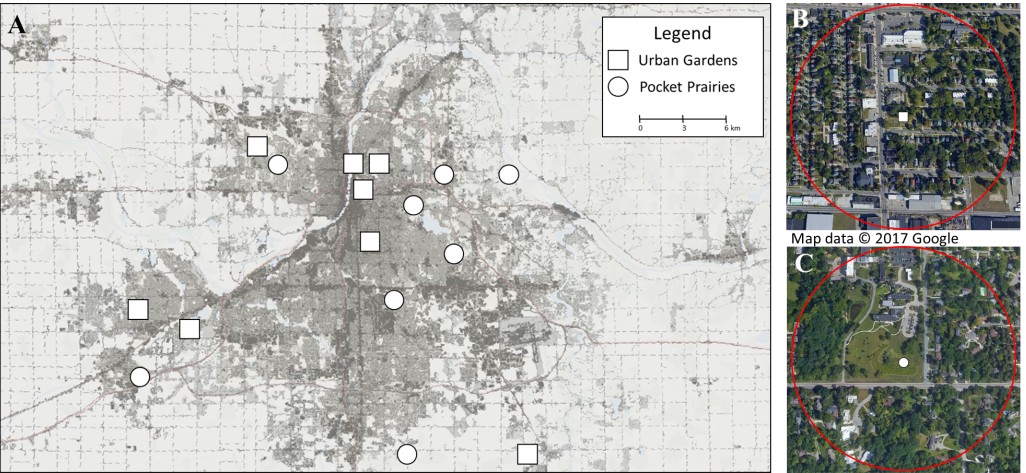

**Figure 1** **Map displaying research sites included in this study.** (A) shows the 16 sites chosen for collection in June and July 2020 in Grand Rapids, Michigan (USA). Sites marked with squares are urban gardens, and sites marked with circles are pocket prairies. The background color shows percent impervious surface, with darker colors containing high densities of impervious surfaces. This map was constructed using the 2016 National Landcover Dataset (NLCD) percent developed imperviousness layer (*Homer et al. 2020*) in ArcGIS, and Google Earth Pro v 7.3.3.7786. The ©2017 Google Imagery was taken on July 30, 2017. For (B) we show an urban garden with a circle surrounding the site to display the 500 m buffer used for calculating impervious surface. For (C), we show a pocket prairie.

Roots, Dominican Sisters of Grand Rapids, Fairway Christian Reformed Church, Rosewood Church, Calvin University, and Cornerstone University.

## Sampling methods

Each site was sampled once per month (June, July) on sunny days with temperatures above 70° F (21.1 °C). Sampling occurred in the morning (8 am–10 am) and afternoon (3 pm–5 pm) on the same day. During each sampling event, multiple measurements were taken near the center of each site in a region that contained visible vegetation representative of the habitat (urban garden, pocket prairie). The temperature and relative humidity were recorded using a digital psychrometer (Model # CECOMINID048683). From the site center, a 15 m transect was marked and the total number of individual flowering plants and their colors (white, yellow, purple) were recorded within 1 m of the transect, as others have done (*Pardee & Philpott, 2014*; *Otoshi, Bichier & Philpott, 2015*; *Burdine & McCluney, 2019b*). We considered inflorescences to be a single flower. To sample the arthropods, 12 large 350 mL bowls (4 blue, 4 yellow, 4 white) and 12 small 175 mL bowls (4 blue, 4 yellow, 4 white) were placed along the transect. Both small and large sampling bowls were used to capture arthropods of varying sizes. Each bowl contained a soap (Dawn) and water mixture to prevent specimens from escaping the bowls. Bowls were set out in the morning and retrieved in the afternoon on the same day. Upon collection, samples were placed into plastic collection containers filled with 70% ethanol for later identification. The collection containers were then labeled and stored at Cornerstone University (Grand Rapids, Michigan, USA).

All arthropods sampled were identified to morphospecies (S1). Arthropod abundance was calculated as the total number of arthropods measured at each site, and morphospecies richness was calculated as the total number of morphospecies present at each site. For the diversity metric, we calculated the Shannon diversity index. We combined arthropod samples from June and July together, and environmental variables between both months were averaged. For temperature and humidity values, we average the measurements taken during morning sampling events.

## Landscape characteristics

The percent of impervious surface for each site was calculated using the 2016 National Land Cover Dataset (NLCD) Percent Developed Imperviousness layer (*Homer et al. 2020*) in ArcGIS v 10.7. The impervious surface value at each site was calculated by placing a 500 m radius buffer around each site, and averaging the impervious surface grid cells from the NLCD dataset. The area, perimeter, and perimeter-area ratio of each site were measured using Google Earth Pro v 7.3.3.7786.

## Statistical methods

All statistical analyses were completed using the statistical program R version 3.6.2 (*R Core Team 2019*). Within this program, we used the "vegan" package to conduct a permutation analysis of variance test (PERMANOVA) on arthropod community composition. Non-metric multidimensional scaling (*meta* MDS) was utilized to display significant associations between arthropod community composition and environmental factors. The "vegan" package was also used to calculate the Shannon diversity index. The correlation function (*cor*) was used to check collinearity between environmental variables (S2). Variables with a correlation coefficient above $r = \pm 0.7$ were considered correlated, and when this occurred one of the variables was removed from the statistical analyses.

Generalized linear models (*GLMs*) were used to compare environmental variables (percent impervious surface, site perimeter-area ratio, temperature, humidity, total flowers, white flowers, purple flowers) against response variables (abundance, diversity, morphospecies richness). For arthropod orders with more than 500 individuals sampled, we used the abundance and morphospecies richness of the order as response variables. We established a list of candidate models that examined each environmental variable separately (S3). From this list of candidate models, the model for each response variable with the lowest AICc value was selected. Models within 2 AICc units were considered equivalent, and these models were combined to determine whether an additive or interactive model was a better fit (S4). When the most parsimonious model for a response metric was within 2 AICc units of the null model, we reported the results of the null. In addition, we used the site type (urban garden, pocket prairie) as an additive variable to evaluate the sites utilized in our study. We chose this process to simplify model selection because additive and interactive models are problematic with model averaging approaches (*Cade, 2015*; *Harrison et al., 2018*). We tested for overdispersion in all models, and made adjustments by altering family distributions when overdispersion was an issue. The distribution families used after testing for overdispersion were negative binomial (total arthropod abundance, Hymenoptera

abundance, Hemiptera abundance, Diptera abundance), Gaussian (diversity), and Poisson (morphospecies richness, Diptera richness, Hemiptera richness, Hymenoptera richness). Assumptions of normality and equal variance were assessed by residual plots, and data transformation were performed when necessary.

## RESULTS

### Summary statistics
A total of 5,468 individuals were collected that represented 14 different taxonomic orders of arthropods and 66 morphospecies. The majority of arthropods sampled (∼88%) were classified into three orders: Diptera (2,604 individuals), Hemiptera (1,156 individuals). and Hymenoptera (1,026 individuals). The orders Araneae (128 individuals), Coleoptera (330 individuals), and Orthoptera (145 individuals) were less abundant, but were represented by at least 100 individuals. The remaining 8 orders contained a combined abundance of 79 individuals (∼1%). Looking at the response variables at the site level, we report the ranges in total arthropod abundance (126–586 individuals), diversity (2.10–2.73), and morphospecies richness (23–39 morphospecies).

The environmental variables we measured also displayed a broad range of values that reflect the conditions of each site. We report the ranges of total flower abundance (23–188 flowers), purple flower abundance (0.5–50.5 flowers), white flower abundance (1.5–70 flowers), humidity (37.7%–65.1%), temperature (21.8 °C–32.1 °C), impervious surface (21.9%–62.3%), and perimeter-area ratio (1.1–5.1).

### Community composition
The PERMANOVA analysis yielded three environmental variables that were significantly associated with arthropod community composition (Table 1). These variables included impervious surface ($F_{1,7} = 2.62$, $p = 0.004$, $R^2 = 0.13$, Fig. 2), white flower abundance ($F_{1,7} = 2.17$, $p = 0.032$, $R^2 = 0.107$, Fig. 2), and humidity ($F_{1,7} = 1.93$, $p = 0.047$, $R^2 = 0.095$, Fig. 2). Site type (urban garden, pocket prairie) was marginally significant ($F_{1,7} = 1.798$, $R^2 = 0.088$, Fig. 2), and the distinction between pocket prairies and urban gardens can be seen in the NMDS plot. Nonmetric dimensional scaling plots indicated that the genus *Chironomus* (midges) were positively associated with impervious surface, and four morphospecies were negatively associated with urbanization: Geometridae, *P. rapae*, Coenagrionidae, and *Camponotus*. Humidity was positively associated with the genus *Ammophila* (thread-waisted wasp).The presence of white flowers appeared to be positively associated with *Lasioglossum, Drosophila, Echenopa,* and *P. japonica*.

### Abundance, diversity, and richness
Here we report results on the most parsimonious model for each response metric (Table 2). Arthropod abundance (AICc = 194.04, $R^2 = 0.39$, Fig. 3A) and arthropod diversity (AICc = −22.23, $R^2 = 0.47$, Fig. 3B) were positively associated with humidity. Arthropod morphospecies richness was negatively associated with the number of purple flowers (AICc = 96.07, $R^2 = 0.39$), but this was equivalent to the null model (AICc = 97.95). Hemiptera abundance was best explained with an additive model of impervious surface and humidity

**Table 1  Results table from the PERMANOVA analysis comparing arthropod community composition with the seven environmental factors.**

| Environmental variable | DF | SS | MS | *F*-value | $R^2$ | *P*-value |
|---|---|---|---|---|---|---|
| Impervious Surface | 1 | 0.216 | 0.216 | 2.623 | 0.129 | 0.004[*] |
| No. White Flowers | 1 | 0.180 | 0.180 | 2.172 | 0.107 | 0.032[*] |
| No. Purple Flowers | 1 | 0.160 | 0.160 | 1.942 | 0.095 | 0.054 |
| Humidity | 1 | 0.160 | 0.160 | 1.934 | 0.095 | 0.047[*] |
| Site Type | 1 | 0.148 | 0.148 | 1.798 | 0.088 | 0.060 |
| No. Total Flowers | 1 | 0.094 | 0.094 | 1.142 | 0.056 | 0.332 |
| Area-Perimeter Ratio | 1 | 0.077 | 0.077 | 0.937 | 0.046 | 0.513 |
| Temperature | 1 | 0.070 | 0.070 | 0.850 | 0.042 | 0.612 |
| Residuals | 7 | 0.577 | 0.083 | | 0.343 | |
| Total | 15 | 1.683 | | | 1 | |

**Notes.**

An asterisk (*) shows a statistically significant results at $\alpha = 0.05$.

(AICc $= 161.21$, $R^2 = 0.57$, Figs. 3A and 3B). Hymenoptera abundance was negatively associated with temperature (AICc $= 145.32$, $R^2 = 0.29$, Fig. 4C), and positively associated with perimeter-area ratio (AICc $= 147.25$, $R^2 = 0.2$, Fig. 4D). Hymenoptera abundance was also explained by an additive model of temperature and perimeter-area ratio (AICc $= 147.32$, $R^2 = 0.36$). The remaining response metrics (Diptera abundance, Hymenoptera richness, Hemiptera richness, Diptera richness) were not statistically different from the null models (Table 2).

## DISCUSSION

Our results reveal important patterns in how arthropod communities are impacted by urbanization. Both abiotic (impervious surface, humidity) and biotic (white flower abundance) conditions influenced overall arthropod community structure, and the composition of morphospecies present. In addition, response variables (abundance, richness, diversity) were also associated with abiotic (impervious surface, temperature, humidity) and biotic (purple flower abundance, perimeter-area ratio) site conditions. This suggests that the stability of arthropod communities and their associated ecosystem services may require multiple intervention points by land managers. The importance of habitat structure (*McIntyre et al., 2001*; *Whitehouse et al., 2004*; *Braaker, Ghazoul & Moretti, 2014*), water availability (*Lagucki, Burdine & McCluney, 2017*; *McCluney, George & Frank, 2018*; *Miles et al., 2019*), and floral resources (*Bennett & Gratton, 2012*; *Burdine & McCluney, 2019b*; *Wilson & Jamieson, 2019*) on arthropods in cities is well documented in the literature. However, the impact of abiotic conditions on Hymenoptera and Hemiptera shows that the impacts of urbanization on these arthropods may be difficult to address. Similar work on bees (Hymenoptera) in cities reveals that increasing floral resources is unlikely to conserve diverse bee communities due to the strong impacts of urban warming (*Hamblin, Youngsteadt & Frank, 2018*).

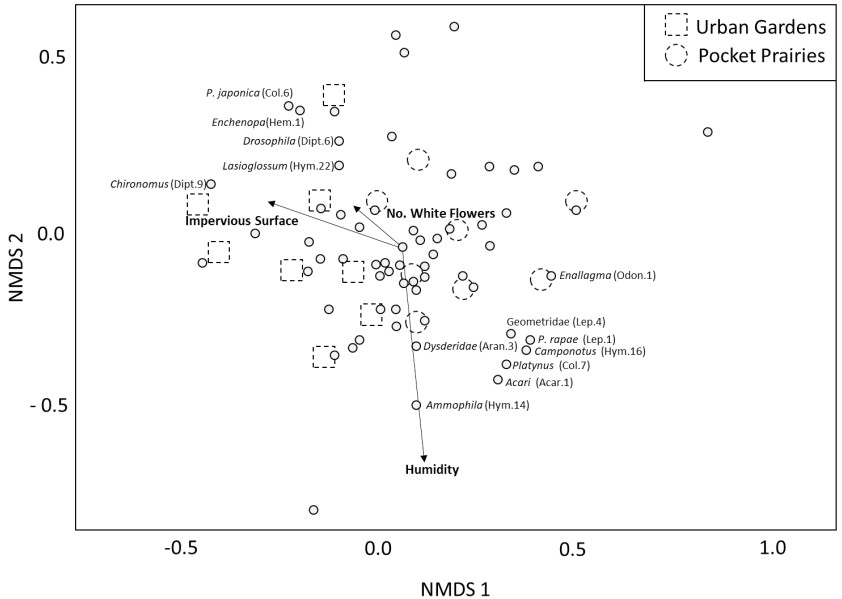

**Figure 2** **Multidimensional scaling plot showing the relative impact that environmental variables have on arthropod community composition.** Significant environmental variables are shown with bolded text and an arrow. Each circle represents an arthropod morphospecies. The names of specific arthropod morphospecies associated with these variables are listed. We display site type by assigning urban gardens with dotted squares, and pocket prairies with dotted circles.

## Humidity and temperature

Temperature and humidity were both important abiotic factors that influenced arthropod communities. Water availability can influence the behavior (*Green, Scharf & Bennett, 2005*) and physiology (*McCluney, Burdine & Frank, 2017*; *Burdine & McCluney, 2019a*) of arthropods, and can limit arthropod populations (*Allen et al., 2014*; *Khaliq et al., 2014*; *Lagucki, Burdine & McCluney, 2017*). Many arthropods that dwell in soil and plant roots require high moisture levels to prevent desiccation (*Bayley & Holmstrup, 1999*), and we know that impervious surfaces in cities change soil moisture by altering hydrology (*Shuster et al., 2005*). Since urban gardens and parks often utilize irrigation systems, these could be important inputs for maintaining abundant and diverse arthropod communities. In addition, repurposing vacant land into raingardens could provide multiple ecosystem services by capturing excess storm water runoff (*Turo & Gardiner, 2019*), and providing habitat for arthropods. Others have suggested that shade trees in urban environments may be important in providing moisture for arthropods. *McCluney & Sabo (2009)* found that crickets increased the consumption of moist leaves to meet their water needs when moisture availability was low. Adding sources of shade and moisture may counteract the negative impacts reduced humidity has on arthropods.

In terms of temperature, many studies have documented the impacts urban warming has on Hymenoptera. Thermal tolerance is often used to explain changes in arthropod community structure (*Youngsteadt et al., 2017*; *Hamblin et al., 2017*; *Miles et al., 2019*), and species-specific responses (*Diamond et al., 2017*; *Burdine & McCluney, 2019a*) across

**Table 2  Results from generalized-linear models (GLMs) displaying the most parsimonious models.** The null model is also included for reference. For each response metric, we considered models within 2 AICc units to be equivalent.

| Response metric | Environmental variable(s) | AICc | $R^2$ | Estimate | SE | T/Z value | P value |
|---|---|---|---|---|---|---|---|
| Abundance | Humidity | 194.04 | 0.39 | 0.036 | 0.010 | 3.543 | <0.001 |
| | Null Model | 199.09 | — | 5.834 | 0.079 | 73.97 | — |
| Diversity | Humidity | −22.23 | 0.47 | −0.015 | 0.004 | 3.666 | 0.003 |
| | Null Model | −14.54 | — | 2.618 | 0.034 | 74.03 | — |
| Richness | No. Purple Flowers | 96.07 | 0.39 | −0.007 | 0.003 | −2.098 | 0.036 |
| | Null Model | 97.95 | — | 3.424 | 0.045 | 75.87 | — |
| Hymenoptera Abundance | Temperature | 145.32 | 0.29 | −0.034 | 0.013 | −2.646 | 0.008 |
| | Perimeter-Area Ratio | 147.25 | 0.20 | 0.117 | 0.060 | 1.950 | 0.051 |
| | Temperature + Perimeter-Area Ratio | 147.32 | 0.36 | −0.027 | 0.013 | −2.070 | 0.039 |
| | | | | 0.075 | 0.057 | 1.303 | 0.193 |
| | Null Model | 147.84 | — | 4.166 | 0.085 | 49.04 | — |
| Hemiptera Abundance | Imperious Surface + Humidity | 161.21 | 0.57 | 0.029 | 0.009 | −3.220 | 0.001 |
| | | | | 0.054 | 0.018 | 3.033 | 0.002 |
| | Null Model | 168.33 | — | 4.276 | 0.161 | 26.52 | — |
| Diptera Abundance | Null Model | 184.93 | — | 5.092 | 0.110 | 46.30 | — |
| Hymenoptera Richness | Null Model | 75.01 | — | 2.404 | 0.075 | 31.98 | — |
| Hemiptera Richness | Null Model | 62.75 | — | 1.447 | 0.121 | 11.93 | — |
| Diptera Richness | Null Model | 64.39 | — | 1.812 | 0.101 | 17.94 | — |

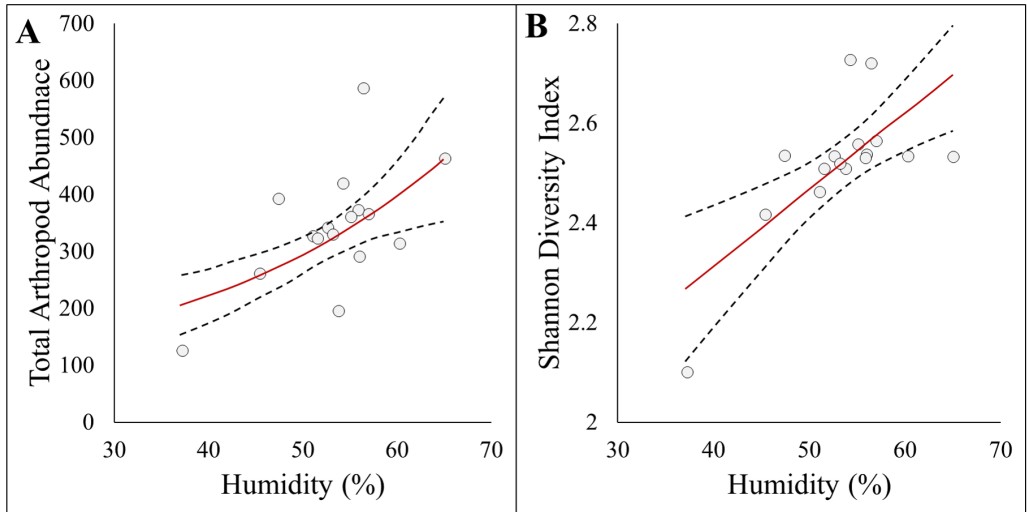

**Figure 3  Panel figure displaying significant associations between response and environmental factors.** Data points represent raw data, and regression lines display non-transformed data. Red lines are fitted regressions from *glm* models, and dashed black lines represent 95% confidence intervals. (A) Positive association between total arthropod abundance and humidity (AICc = 194.94, $R^2$ = 0.39). (B) Positive association between Shannon diversity index and humidity (AICc = −22.23, $R^2$ = 0.47).

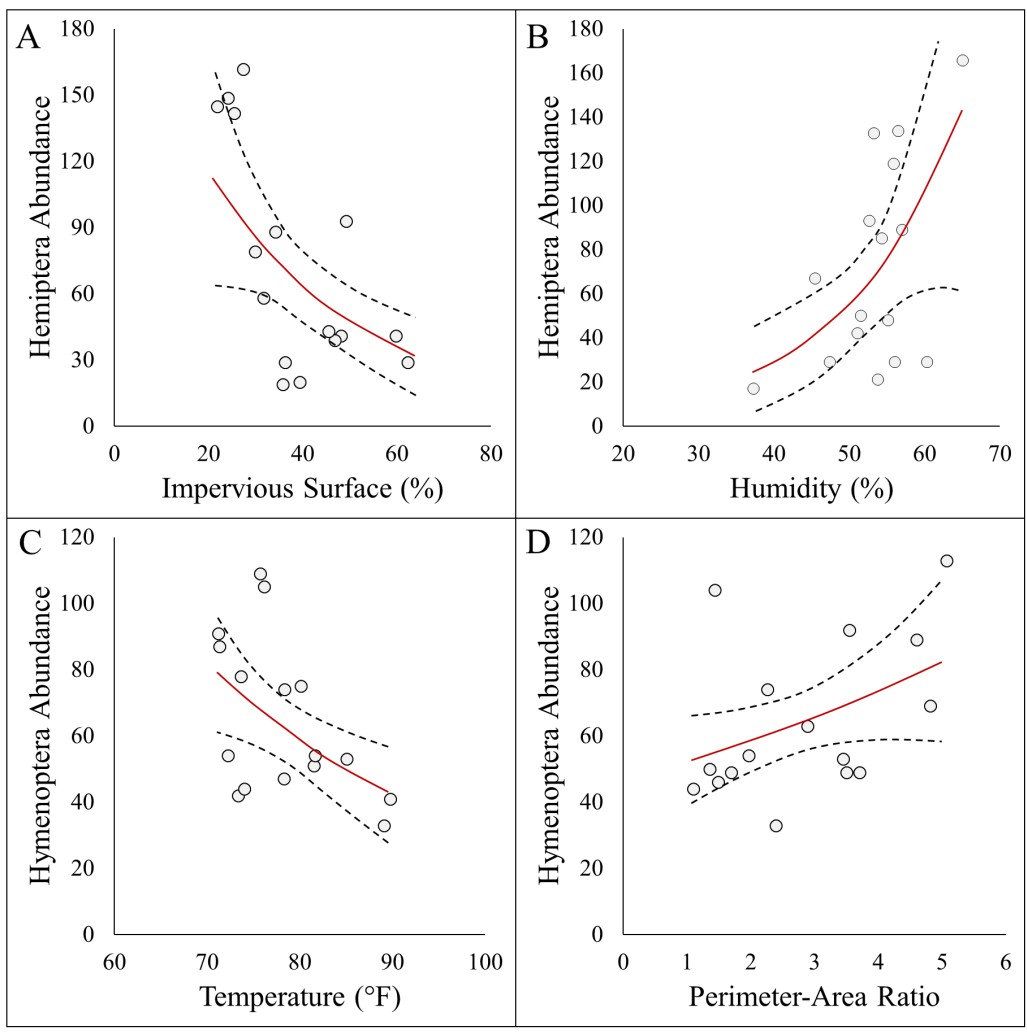

**Figure 4** **Panel figure displaying significant associations between the abundances of Hymenoptera and Hemiptera and environmental factors.** Data points represent raw data, and regression lines display non-transformed data. Red lines are fitted regressions from *glm* models, and dashed black lines represent 95% confidence intervals. Hemiptera abundance was best explained by an additive model (AICc = 161.21, R2 = 0.57) that showed a (A) negative association with impervious surfaces, and (B) positive association with humidity . Hymenoptera abundance showed a (C) negative association with temperature (AICc = 145.32, $R^2 = 0.29$), and (D) positive association with site perimeter-area ratio (AICc = 147.25, $R^2 = 0.20$).

urbanization gradients. For instance, *Hamblin, Youngsteadt & Frank (2018)* found sharp declines in bee abundance across an urbanization gradient, and temperature was the dominant variables in explaining these changes. Therefore, it is not surprising that we detected declines in Hymenoptera with temperature. We did identify a positive association between white flower abundance with *Lasioglossum* (sweat bees), but it is notable that overall floral resources availability had no influence on Hymenoptera. This suggests that simply planting more flowers may not impact Hymenoptera unless interventions addressing temperature are addressed.

## Site structure

Impervious surface was a strong driver of arthropod community composition. In particular, we observed declines in Hemiptera (true bugs) with increasing impervious surface. In general, arthropod declines are common in cities (*Ahrné et al. 2009*, *Fenoglio, Rossetti & Videla, 2020*), as impervious surfaces replace natural vegetation cover. However, even at our most urban site (62.3% impervious surface) there is still available habitat for arthropods to utilize. *Knop (2016)* documented the broad homogenization that true bugs have experienced in cities, and the loss of host plants may underlie this homogenization. We did find that *Enchenopa* (tree hoppers) were positively associated with white flower abundance, but this was the only Hemiptera morphospecies associated with floral resources. In addition, a weak negative relationship between morphospecies richness and purple flower abundance was found. These results suggest that specific types of floral resources are required for some arthropod taxa, and that simply increasing overall floral resource availability may not be enough. Additional research is needed to investigate whether targeted plant additions at high impervious sites can counter the negative impacts of impervious surfaces, and how much space is required to make an impact.

Urban greenspaces are often small, and embedded within an urban matrix that promotes generalist arthropod species (*Gaublomme et al., 2008*). Small urban sites generally do not contain the core habitat needed to maintain diverse arthropods (*Christie, Cassis & Hochuli, 2010*), and may be more prone to edge effects. Studies have shown that urban greenspace patch size is a correlate of communities composition for ants (*Uno & J. Cotton, 2010*) and natural enemies (*Burkman & Gardiner, 2014*). Our results suggest that Hymenoptera are also influenced by site structure, as Hymenoptera abundance increased with perimeter-area ratios. Strategies to increase the size and connectivity of urban greenspaces may be important in conserving arthropods, particularly low-mobility arthropod species (*Braaker, Ghazoul & Moretti, 2014*). Sources of new greenspace habitat is available in vacant lots (*Gardiner, Burkman & Prajzner, 2013*), green roofs (*Braaker, Ghazoul & Moretti, 2014*), or backyard gardens (*Pardee & Philpott, 2014*), amongst others. Increasing the quantity and quality of urban greenspaces not only benefits arthropods, but may enhance the quality of arthropod-mediated ecosystem services like pollination and biological control (*Sánchez Domínguez et al., 2020*). As cities continue to grow and expand into the surrounding peri-urban environment, city planners should consider leaving patches of non-impervious habitat to maintain diverse arthropod assemblages in cities.

Future research should be conducted on the specific ecosystem services provided by individual taxa to better understand how urbanization influences the delivery of these services. There is also a strong need for manipulative studies to investigate whether the mitigation of urban effects (increasing greenspace size, adding shade trees, irrigation) has a positive impact on arthropod communities. Many of the abiotic conditions we documented could also be explained by loss of greenspace habitat, and manipulative studies would provide more precise conclusions. These types of studies would provide critical pieces of information that land managers and urban dwellers could implement on their properties. This study is limited in its ability to provide clear recommendations on

the importance of flower species, and future studies to identify floral species in lieu of using floral colors as environmental factors should be necessary. As many arthropods utilized specific host plants, this information would produce a more applicable recommendation to land managers.

## CONCLUSIONS

Our results show that both abiotic and biotic conditions are important in the structure of arthropod communities, and plans to conserve arthropods in cities should consider both. The ecosystem services arthropods provide to humans living within urban environments are valuable. As urban expansion continues, city planners should consider steps to mitigate the impacts of urban heat islands and altered water availability on arthropods and their associated ecosystem services. Increasing the availability of urban greenspaces or repurposing existing urban lots may be important strategies for maintaining diverse arthropod assemblages in urban ecosystems.

## ACKNOWLEDGEMENTS

We thank the local cities and townships, nonprofit organizations, educational institutions, and faith-based groups for providing access to study sites used for collection. We also thank Cornerstone University for providing us with the facility and resources needed to complete this research. And to Dr. Rob Keys for supplying additional materials and advising throughout the study. In addition, we thank the reviewers and editor of this manuscript for providing valuable feedback that tremendously increased the quality of the manuscript.

### Funding
This work was supported by Cornerstone University. The funders had no role in study design, data collection and analysis, decision to publish, or preparation of the manuscript.

### Grant Disclosures
The following grant information was disclosed by the authors:
Cornerstone University.

### Competing Interests
The authors declare there are no competing interests.

### Author Contributions
- Garrett M. Maher and Justin D. Burdine conceived and designed the experiments, performed the experiments, analyzed the data, prepared figures and/or tables, authored or reviewed drafts of the paper, and approved the final draft.
- Graham A. Johnson conceived and designed the experiments, prepared figures and/or tables, authored or reviewed drafts of the paper, and approved the final draft.

### Field Study Permissions

The following information was supplied relating to field study approvals (i.e., approving body and any reference numbers):

We received verbal permissions to collect samples from the following institutions: City of Grand Rapids, City of Hudsonville, Township of Ada, Township of Gaines, Creston Neighborhood Association, Blandford Nature Center, New City Neighbors, Urban Roots, Dominican Sisters of Grand Rapids, Fairway Christian Reformed Church, Rosewood Church, Calvin University, and Cornerstone University.

### Data Availability

The raw data and the R Code are available in the Supplemental Files.

### Supplemental Information

Supplemental information for this article can be found online at http://dx.doi.org/10.7717/peerj.12818#supplemental-information.

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
