# Peer review of "Impervious surface and local abiotic conditions influence arthropod communities within urban greenspaces"

_PeerJ, doi:10.7717/peerj.12818_

## Round 0.1 · original submission · Major Revisions

Dear Dr. Maher and colleagues:

Thanks for submitting your manuscript to PeerJ. I have now received three independent reviews of your work, and as you will see, the reviewers raised some concerns about the research. Despite this, these reviewers are very optimistic about your work and the potential impact it will have on research studying arthropod communities within urban greenspaces. Thus, I encourage you to revise your manuscript, accordingly, taking into account all of the concerns raised by the reviewers.

The main concern raised by all three reviewers is the level of taxonomy for arthropod assignment (too low and vague). I agree with the reviewers, and incorporating a higher-level taxon assignment will increase the relevance of your work and the statistical significance.

There are many other concerns pointed out by the reviewers, and you will need to address all of these and expect a thorough review of your revised manuscript by these same reviewers.

Please be sure to include the missing references identified by the reviewers and make sure the figures and tables are clear and report all information such that all analyses may be repeatable.

I look forward to seeing your revision, and thanks again for submitting your work to PeerJ.

Good luck with your revision,

Best,

-joe

·

Basic reporting

see below

Experimental design

see below

Validity of the findings

see below

Additional comments

Dear authors,
Thank you for the opportunity to read and review your text. I read the paper with great interest and I think that the topic is very important. In particular, I appreciate that you have taken a multi-taxon approach to your work, which is extremely important for understanding biodiversity dynamics along rural-urban gradients. I understand that identifying such a large number of different orders is very difficult and time-consuming, but I think that the importance and significance of your study would be much greater if it were possible to identify at least some orders further down to species. In the current version, unfortunately, too much significance is lost for me, as the environmental effects are only analysed in relation to a very rough taxonomic level. Symptomatic of this is, for example, the NMDS (Figure 2), which only shows the arrangement of different orders along the gradients. What is really exciting, however, is whether certain species communities develop within the orders and whether there is an effect of urbanisation.
I am sorry that this criticism means that I cannot recommend the text for acceptance. Overall, the methods are very well chosen, the study design is very good and the flow of thoughts is also very good. I would therefore encourage you to identify the species in at least some orders and then repeat the analysis. If you do this, you could still consider the different management of the plots (I had understood it to mean that the plots differ in this respect). If a further determination of the species is not possible, it would be a possibility to work with morphospecies in order to have a better value for diversity. Taking the number of taxonomic orders present at each site as a measure of species diversity is problematic because it is very inaccurate and not a measure of actual species diversity.
I wish you much success and would be happy to review a new version of the text.
Best regards,
Sascha Buchholz

Reviewer 2 ·

Basic reporting

This study investigates the arthropod communities within different urban greenspaces across metropolitan Grand Rapids, Michigan (USA). The manuscript is well written and the general idea of the work is good, I enjoyed reading it. However, I found some issues that are of concern. The main observation is the use of level order to determine the species richness and diversity. I am afraid it is difficult to obtain conclusions regarding biodiversity with such a broad level of identification. For abundance, I think it is not a problem, but it is for richness. Authors should give a proper explanation of why they did that when available keys and specialists are in the USA. Otherwise, they should be very careful with the scope of their conclusions and mention this point as a limitation of the study. On the other hand, I am worried that the authors did not distinguish in the models between the two types of green areas used. Urban gardens and pocket prairies may be different in terms of plant species richness, plant origin, management, etc, so if authors did not measure any of these variables at least they must include the type of site into the models.
In the Introduction there is no mention of the possible effects of flower availability on insects, a variable included then into the models. One would expect that will be important for pollinators, of course, but what about other groups? Why also did you explore the effects of different colors? Please explain and give predictions also about that. In addition, there is no reference to temperature and humidity in terms of previous evidence on insects in urban areas or what you could expect regarding those variables in your study system. Please add some background on that which may help to formulate your predictions. The work of Fenoglio et al 2021 could help to give you a broader picture of the inter-relations between different urbanization drivers.

Fenoglio, M. S., Calviño, A., González, E., Salvo, A., & Videla, M. (2021). Urbanization drivers and underlying mechanisms of terrestrial insect diversity loss in cities. Ecological Entomology. DOI: 10.1111/een.13041

Experimental design

Several issues in Material and Methods should be addressed:
Line 103: Please characterize a bit more the two types of sites considered
Line 104: are there familiar or community gardens?
Lines 117-118: Which values did you use of temperature and humidity in the analyses? What was the average mean daily temperature during the sampling period? Please clarify
Line 118: Regarding the 15m transect, how did you choose the starting point, you went always to the north/south, please add more information.
Line 119: You measured only the number of total flowers, but not plant species richness which is known to affect insect diversity. Why that? Please clarify
Line 119: You have considered four colors of flowers, please explain or cite a reference to justify that selection
Lines 120-121: Why did you use two sizes of bowls? Please explain
Lines 128: You should provide some kind of justification for estimating species richness at the order level. Besides, in methods, you did not mention anything about diversity (how did you calculate it for example). I strongly suggest being consistent along with your manuscript with the response variables used as well as with the explicative variables, from the introduction to the discussion.
Line 153: Here you said that you used as a response variable the abundance of two orders, why that? Please explain better
Line 155: The models showed in S3 to S7, which response variable have? It is not clear to me what did you do. Did you do that to select the variables entering the model? why? In addition, you could condense all these separate tables into one larger in order to have fewer supplementary files.

Validity of the findings

Results
Lines 189-190: Now it seems that you considered also Diptera abundance but you did not mention it in methods. Please, put this part in the methods sections and justify it.
Fig 3D: Please change the name of the “y” ax to arthropod richness

Discussion
In general, I found you did not discuss some parts of the results. For example, there are no almost discussions related to temperature effects on overall insect abundance and Hymenoptera abundance (I see in references the work of Dale & Frank, 2017 but it is not mentioned in the text).
Urban Ecosystem, 21, 419–428.
In addition, when you talk about the importance of flower additions for insect abundance (lines 218-220), please be careful in the way you explain that since you found that the number of flowers had no effect on it. Besides, you did not discuss anything about the effects of different variables on insect composition, for instance, the close relation between humidity and Dermaptera or Hymenoptera and white flowers.
You should mention the caveats of the study and the scope of the interpretations of your results if you insist on maintaining diversity at the order level. You possibly will be underestimating the effects of any of the other variables, beyond the impervious surface, if you consider such a level of resolution in your taxonomic identification.

Additional comments

I think the article could improve a lot after considering all my commentaries.

Reviewer 3 ·

Basic reporting

I have reviewed the article entitled “A survey of above-ground arthropod communities within urban greenspaces in Grand Rapids, Michigan (USA)”. Maher and collaborators investigated insect communities in urban gardens and prairies and their responses to environmental factors related to urbanization and flower availability. While the topic is not entirely novel and there are several studies addressing similar questions, I think it can be a valuable contribution to improving our understanding of community-level responses to urbanization. Nevertheless, I have found many important issues that need to be addressed in order to improve the data analyses and provide clear communication of their results. In my opinion, the article could be substantially improved after revising all the listed aspects. In the following three sections, I outline my major suggestions, whereas in “general comments to the author” I listed several minor suggestions.

English writing is good and clear in most of the manuscript. The context provided in the introduction is supported with appropriate references and generally covers the needed topics. Perhaps the paragraph on lines 51-68 condenses a lot of information, but it is not entirely clear. Maybe the first part, regarding impervious surfaces and their effects on the abiotic conditions of cities could be integrated with the first paragraph and the second could be focused on biotic homogenization?

Raw data is fully provided as supplementary tables. Finally, all the figures are justified and of good quality, though I have some suggestions for all of them.
-Figure 1: It would be good to add a panel showing one site as an example, with a landscape circle around the study site to clearly visualize the impervious cover in one of the sites. Perhaps adding a photo of one garden and one prairie in Figure 1 on a separate panel would help readers to get a better idea of how the two types of sampled habitats.
-Figure 2: I think that plotting the study sites in the NMDS would help to visualize how different were your sites if gardens and prairies had particular environmental conditions, and if the insect orders were related to particular sites as well.
-Figure 3: the legend from Fig. 3 needs more detail. State clearly that you are showing raw data points and what do the lines represent. Are those the fitted lines from the GLMs? Also, adding the confidence intervals for those fitted relationships would be nice.

Experimental design

-I believe that the research topic is relevant and fits well within the scope of the journal. At the same time, the main research question and the specific questions addressed by the authors are specified, although the hypotheses and predictions need some additional work.
Firstly, not all investigated drivers are described in the introduction. The focus is on the gradient of impervious, but adding a small paragraph with more details on the remaining environmental drivers would be helpful to understand their design. In connection to this, the hypotheses and predictions are a bit vague and the authors’ expectations regarding drivers other than the impervious surface would be interesting. I think the choice of the variables is logical and that the authors had previous ideas about how them would affect insect, so it would be good if they described them in the hypotheses. On another hand, the choice of particular flower colours as predictor variables needs to be justified as well. Was that based on a priori hypotheses or literature? Or was the decision taken based on the abundance of flowers?
-A relevant issue, in my opinion, is the study sites. Gardens and prairies are used as replicates but they can certainly be different. I imagine that prairies and urban gardens are structurally very dissimilar and are subjected to different levels of management. Therefore, I think habitat type should be included as an additional predictor in the models to account for this possible difference. Note that this is also interesting, as the authors could determine if one of these urban green spaces has a higher potential for supporting insect communities. Finally, I believe pocket prairies need to be defined clearly. I am not familiar with the term and the text gives only a slight idea of what they are.
-Another important thing is that the authors use the number of orders as a proxy of richness and to calculate indices of diversity and evenness. I believe that more caution should be taken here. Despite that there is a clear variation in the number of orders, I do not agree that this can be called “species richness”. I am not familiar with studies using order richness as the response variable and calling this species richness, not even in those studies using family richness. Thus, I think that throughout the text, if the authors decide to keep order richness, the variable should be called like that (and community/order diversity and evenness, perhaps?). And if there are studies published using a similar method, it would be good to cite them in methods. Furthermore, I expect at least a brief mention of this limitation in the discussion as well.
-L 146-150: why did you measure collinearity between response variables? I agree with the method regarding environmental variables, but I don’t think response variables need to be reduced in this way, as they are not evaluated within the same model. Even if Diptera and total abundance are correlated, they could respond differently to environmental variables, so I think you should use Diptera abundance as a response variable as well.
-Statistical analyses: Why not performing models with more than one environmental variable? Interactions between variables probably cannot be analysed due to the reduced number of replicates, but additive models with at least combinations of two environmental variables could be tested, as both structural and climatic variables could affect your response variables conjunctly (see Fenoglio et al., 2021 for a revision of the combined effects of multiple drivers). In connection with this, I would include habitat type as an additional predictor as mentioned above. Moreover, the results of additive models including, for example, both impervious cover and flower availability may provide a clue about the effects of both factors discussed in lines L213-224.
The R code provided in the supplementary material suggests that you had a full model with all the environmental drivers, but this is not reported and the methods suggest that only univariate models were tested. Also, the code and text suggest you used GLMs, but the family (i.e., error distribution) is not specified. Was it a Poisson distribution for order richness and abundance? Was overdispersion not a problem with abundance data? Models for diversity were GLMs or LMs with a Gaussian distribution? This needs to be described in more detail.
In connection with statistical analyses, why did the authors average the June and July samples instead of using separate samples from each month and using mixed models? A lot of variability in the data linked to climatic variations (and/or flower availability) could be lost, reducing the probability of finding relevant effects of these variables.

Validity of the findings

I think the replication is acceptable and allows the authors to test for the investigated factors. The data are robust but the taxonomical level of identification should be justified, and the statistical analyses should be revised (see comments in the previous section).

The conclusions section is mostly describing which type of studies are needed in the future, so I think it needs to be reshaped. These observations could be included in a paragraph addressing the taxonomical limitations of the study, whereas the conclusion section should focus more on the interpretation and relevance of their result. I encourage the authors to re-analyse their dataset following some of my previous suggestions and I expect this section to change at least partially.

Additional comments

Minor comments:
-This is just an opinion, but the title is not very attractive and sounds rather local. I think switching to a title stating the main results would be much more interesting and catchy.
-L 80: the end of the sentence is not completely clear to me. Are you stating the previous research (the references from that sentence?) suggest that greenspaces benefit people and arthropods? It reads as if performing research on green spaces provides the benefit, but I do not see the link.
-L 93: perhaps here it could be specified that impervious surfaces were measured at the landscape scale. Some studies include this variable on a rather local scale, so it is good to indicate which approach you took from the beginning.
-L 106-108: I don’t see why the permissions for using those discarded 4 sites should be mentioned. Instead, I would only leave the minimum distance that we selected to maximize spatial independence.
-L 114: considering the international readership of the journal, I suggest at least add the temperature in °C.
-L 153-154: were Hymenoptera and Hemiptera (please use uppercase letters when writing the Latin order name) chosen based on their abundance? If so, please indicate the minimum abundance that you considered to make analyses on separate orders.
-L 207-208: in my opinion, this sentence would read better if rephrased as “shows that the impacts of urbanization may be difficult to overcome for some taxa” or “shows that for some taxa the impacts of urbanization may be difficult to overcome”.
-L 214-215: mentions to figures are not common within the discussion and, since you only refer to these two examples, I believe it is better to remove them.
-Lines 235-237: indeed. There are evidence for this that you could mention in this sentence. For example, Sanchez Dominguez et al. (2020) found that landscape-scale grassland cover increased predation rates.
-L 242-254: I think more discussion on temperature effects (i.e., heat island effect) is needed here. Despite that most studies working on the urban heat islands are focused on herbivorous insects and not on community-level responses, McGlynn et al. (2019) is a recent example of how temperature affects a highly diverse insect group.

References cited:
-Fenoglio, M. S., Calviño, A., González, E., Salvo, A., & Videla, M. (2021). Urbanisation drivers and underlying mechanisms of terrestrial insect diversity loss in cities. Ecological Entomology.
- McGlynn, T. P., Meineke, E. K., Bahlai, C. A., Li, E., Hartop, E. A., Adams, B. J., & Brown, B. V. (2019). Temperature accounts for the biodiversity of a hyperdiverse group of insects in urban Los Angeles. Proceedings of the Royal Society B, 286(1912), 20191818.
- Sanchez Domínguez, M. V., González, E., Fabián, D., Salvo, A., & Fenoglio, M. S. (2020). Arthropod diversity and ecological processes on green roofs in a semi-rural area of Argentina: Similarity to neighbor ground habitats and landscape effects. Landscape and Urban Planning, 199, 103816.

---

## Round 0.2 · Major Revisions

Dear Dr. Maher and colleagues:

Thanks for revising your manuscript. The reviewers are optimistic with your revision (as am I). Great! However, there are some remaining concerns to address (per reviewers 2 and 3). Particular attention to the statistics should be made.

Please address these ASAP so we may proceed with your work.

Best,

-joe

·

Basic reporting

Thank you for the revision of the text. I am very glad that it was possible to make further species identifications and thus significantly increase the information content and informative value of the study. It is still a very large consideration of urban biodiversity but nevertheless the morphospecies approach succeeds very well in making an interesting contribution to urban ecological biodiversity research. In my opinion, the study is now an interesting contribution that should definitely be published. The statistical models have clearly gained in precision and the results and statements are now easier to understand.

Experimental design

See 1.

Validity of the findings

See 1.

Additional comments

no comments

Reviewer 2 ·

Basic reporting

The manuscript has improved a lot since the authors considered most of the aspects that reviewers pointed out. I really appreciate their effort to
Identify to a lower taxonomic resolution the species found. However, there are still some issues (especially related to statistical analyses) that need to be to be revisited or better justified. I consider that the manuscript will be publishable after the authors consider these comments.

Experimental design

The main problem I see is related to the construction of the statistical models. Previously, one of the reviewers has observed and asked why doing univariate models instead of additive ones. The authors argued that by parsimony they decided still work with univariate models. However, for me is not a fully reasonable explanation since biologically is more probable that two variables like humidity and temperature act in an additive way than separately. If authors tested those models, as they mentioned in the rebuttal letter, why they do not include those results as well? Besides the problem of multiple hypothesis testing is that there is always a chance that what the result considers true is actually false (Type I error), thus some kind of correction should be applied (perhaps Bonferroni is too conservative, but there are others). In addition, I do not understand why they tested multicollinearity between independent variables if then they used univariate models. This is not consistent, please correct it. In addition, there is a mistake in the term used in line 160: the correlation test was used to “check” possible multicollinearity problems not to “reduce” as they said. Please, in case you decide to keep the models the way they are, I strongly suggest adding a reference to support your model construction and selection.
On the other hand, the authors mentioned that quantified the number of flowers of different colors but in the analyses, red and orange flowers did not appear and the reader does not know why. Please, clarify.
Regarding overdispersion testing, is not clear which adjustments they did…did they change the distribution? To a quasipoisson or negative binomial? Please better explain

Validity of the findings

The validity of the findings depends on a clear justification of the statistical method used or instead the use of a proper one that consider several variables in just one model.

Reviewer 3 ·

Basic reporting

I thank the authors for their work on the revised version of the manuscript. Several sections have been improved and most of the comments were addressed. The main change is that arthropods were identified to morphospecies, which allows for a more detailed analysis. Also, I see clear improvements in the introduction, the delineation of their goals, and the description of the study sites. Nevertheless, I think that a few major things remain unclear and I also listed several minor comments. I am sorry that I cannot recommend your manuscript for publication in its current state, but I am convinced that these changes will improve the quality of your work. I look forward to reading the revised version of your paper.

Major comments:
-Regarding analyses, I am still not fully convinced about the approach they used. As mentioned in the first revision, fitting models with more than one predictor is a standard practice in ecology. If the authors are convinced that fitting simple univariate models is better for a particular reason, I think this should be clear in the methods and supported by a reference. At the same time, the rebuttal letter indicates that when equivalent models were detected, an additive model was tested but this is not mentioned in the manuscript or reported in the results. Therefore, please expand this section by stating exactly what you did and report the models in your tables. Even if the additive models are not better, I would still want to see their estimates and details. In connection with analyses.
In the revised version, the authors have included site type as a predictor after the suggestions, but this was also done in a univariate model. This can be seen in the tables, but not in the R code, please check that. My point is that site type is not an urbanization/habitat variable as the remaining factors, but a variable that is inherent in your study sites. Thus, I believe that site type should be always included as a predictor in addition to any other variable, in order to contemplate the design of your study.
Finally, the authors stated that they used GLMs, but the R code shows that for both abundance (log-transformed) and diversity, LMs with Gaussian distribution is used. Was morphospecies richness analysed using a Poisson distribution? As the R code includes only the first two response variables, this is not clear. Also, why use log-transformed data when negative binomial GLMs are good for dealing with abundance data? Please explain the distributions used for each response variable and justify the transformation for abundance.
-In connection with my comment on study sites, Permanova showed that this variable had a marginally significant influence on the community composition and I think it should also be included in the final model. I repeat my original comment that including the study sites in the NMDS using the same symbols as in Fig. 1 will help to better understand how different was the composition between gardens and prairies.
-In the results section, table S2 shows that morphospecies richness was negatively linked to purple flowers, even though this model was similar to the null model in terms of AICc. Even if this means that the support from the data was weak, I think this should be reported and discussed as well.
-The title and discussion emphasize that abiotic conditions are influencing the responses of insects, but keep in mind that impervious cover effects can reflect the negative effects of losing green habitats. In other words, can you be sure that this is entirely an abiotic driver, or could the low availability of green spaces (biotic factor) explain your result? Most of the results are linked to temperature and humidity, but still, you could be more cautious about that response to impervious cover.

Minor comments:
-L 52: perhaps obligatory is not the best word to describe these ecosystem services? Essential or needed?
-L 76-78: McGlynn et al worked with only one diverse family, so I think this sentence should not be so general. Also, the specific group studied by Andrade et al is missing.
-L 106-107: here it should be clearer that 8 gardens and 8 prairies were sampled, it is not completely clear in the text.
-Table S2 is hard to read due to the line formatting. I suggest using horizontal lines or different background colours to separate different response variables.
-Table S2: what do the estimate of site type reflects? Were gardens the reference level?
-L 188-190: by individuals you mean flowers? Please modify this, individual flowers are not directly linked to individual plants.
-In connection to my previous comment, which criteria did you use for counting inflorescences? Were they counted as one flower?
-L 193 and 204: I think the word results is not needed for these two sub-headings.
-L 201: it should be white flowers.
-Table 2: following the same logic as in the response variable shown here, I think that for the remaining response variables the null model should be listed if it was the best one.
-L 235-237: this point highlights that garden could be more humid than prairies and thus site and humidity should be included in the same model to disentangle their influences.
-L 263: underlie.
-Fig. 3 and 4: are the plotted regression lines the ones fitted in the models, or simple lines added to the raw, untransformed data? I think this should be clearer in the legend and I insist that adding confidence intervals would be better to visualize the responses of arthropods.

Experimental design

All comments are included in basic reporting.

Validity of the findings

All comments are included in basic reporting.

---

## Round 0.3 · Minor Revisions

Dear Dr. Maher and colleagues:

Thanks for again revising your manuscript. The reviewers are super optimistic with your revision (as am I). Great! However, there are a few remaining concerns to address (again, per reviewers 2 and 3). These are very minor in scope.

Please address these ASAP so we may move forward with publishing your work.

Best,

-joe

Reviewer 2 ·

Basic reporting

This new version of the manuscript has improved a lot. I appreciate the effort of the authors. I suggested only minor changes below detailed.

Abstract
-First sentences: Please use a synonymous in order to not repeat so many times the word decline
-Line 18. replace for few studies in cities
-Lines 19-22. Reframe these two sentences, they are repetitive: In this study, we surveyed arthropod communities within a medium-sized, growing city. We investigated arthropod communities (abundance, richness, diversity, community composition) within urban greenspaces across metropolitan Grand Rapids, Michigan (USA).
-Line 23. Be more specific about which environmental variables are you considering
Introduction
Line 64. where it says “as specialist species are replaced…” should be added "native"
Line 69. add an “s” to animal
Line 98. Please be more precise regarding which environmental variables you evaluated
Material & Methods
Line 110. Where it says “but often contained non-edible flowers” shouldn´t be vegetables?
Line 151. Correct to: The percent of impervious
Line 152. It is rare the expression “percent developed imperviousness layer”, revisit it
Line 168. It should be GLMs
Lines 172-173. I am still not quite convinced about first testing each environmental variable separately, but I agree that this section has improved a lot.
Line 180. You said “made adjustments by altering family distributions”, please be more precise referring to which distribution did you used
Discussion
Lines 234-236. Please re-write this sentence, it is not clear what did these variables have an effect on?
Line 243. Correct to: impacts of urbanization
Line 313. Replace: “we recommend future studies” to “future studies to identify floral species in lieu of using floral colors as environmental factors should be necessary” in order to be not repetitive with the term recommend

Experimental design

detailed before

Validity of the findings

detailed before

Additional comments

detailed before

Reviewer 3 ·

Basic reporting

Thank you for revising your article and following the comments made on the previous version. I believe the article has improved considerably and it is much closer to, in my opinion, being accepted for publication. The methods and the figures are now clearer. I only have a few minor comments to report.

-L 175: the model selection description mentions that models within 2 AICc units were considered equivalent, but for several groups (especially for morphospecies richness) the null model and other models had similar AICc values, but only the null model is reported in Table 2. I believe that either every model with the deltaAICc = 2 should be reported there or that sentence in methods should clarify that if the null model had the lowest AICc value the selection process was different.
-L 175-179: thanks for including the additive and interactive models in the supplementary material. Even though I agree that interactive models can be problematic when doing model averaging, additive models are usually ok and the authors are not doing model averaging in their work, so the statement may not be necessary. Personally, I would have analysed the data differently by fitting additive models including all the possible combinations of predictors, but given that the authors provided suitable references for their choice I can accept it if the editor approves it.
-L 217: perhaps a better sub-heading could be used for this section. One that is not focused on the method you used, but on the responses of arthropods to the factors you studied.
-L 234-236: with univariate responses you mean the responses of each order? Please clarify. Also, the sentence could be better connected with the previous one.

Experimental design

See 1.

Validity of the findings

See 1.

---

## Round 0.4 · accepted · Accept

Dear Dr. Maher and colleagues:

Thanks for revising your manuscript based on the concerns raised by the reviewers. I now believe that your manuscript is suitable for publication. Congratulations! I look forward to seeing this work in print, and I anticipate it being an important resource for groups studying arthropod communities within urban greenspaces. Thanks again for choosing PeerJ to publish such important work.

Best,

-joe